# Revealing the hidden carbon in forested wetland soils

Anthony J. Stewart [1] ✉, Meghan Halabisky[1], Chad Babcock [2], David E. Butman [1], David V. D'Amore[3] & L. Monika Moskal [1]

Inland wetlands are critical carbon reservoirs storing 30% of global soil organic carbon (SOC) within 6% of the land surface. However, forested regions contain SOC-rich wetlands that are not included in current maps, which we refer to as 'cryptic carbon'. Here, to demonstrate the magnitude and distribution of cryptic carbon, we measure and map SOC stocks as a function of a continuous, upland-to-wetland gradient across the Hoh River Watershed (HRW) in the Pacific Northwest of the U.S., comprising 68,145 ha. Total catchment SOC at 30 cm depth (5.0 TgC) is between estimates from global SOC maps (GSOC: 3.9 TgC; SoilGrids: 7.8 TgC). For wetland SOC, our 1 m stock estimates are substantially higher (Mean: 259 MgC ha$^{-1}$; Total: 1.7 TgC) compared to current wetland-specific SOC maps derived from a combination of U.S. national datasets (Mean: 184 MgC ha$^{-1}$; Total: 0.3 TgC). We show that total unmapped or cryptic carbon is 1.5 TgC and when added to current estimates, increases the estimated wetland SOC stock to 1.8 TgC or by 482%, which highlights the vast stores of SOC that are not mapped and contained in unprotected and vulnerable wetlands.

Conserving Earth's carbon-rich ecosystems is critical in order to meet the goals of balancing carbon sources and sinks for the Paris Climate Agreement[1]. Among ecosystems with high carbon stocks, inland freshwater wetlands and peatlands contain greater than 30% of the global total soil organic carbon (SOC) stock of 1500–2400 PgC but only cover approximately 6% of the land surface[2–4]. However, below the global scale, wetland SOC mapping is considerably more uncertain due to poor spatial representation. Estimates of wetland SOC stocks often rely on coarse resolution mapping and broad scale inventories that omit many wetlands outside of large homogenous wetland complexes such as peatland plateaus in the high latitude northern hemisphere (>60°)[5,6]. In the more heterogenous, complex terrain of mid-latitude temperate forested regions (30°–60°), wetlands still disproportionately contribute towards terrestrial carbon storage compared to upland areas, but are difficult to map and can occur subtly within a forested landscape and remain hidden under the canopy[7]. This temperate wetland area has been a frequent target of land use

conversion to agriculture and urban land uses contributing to the recently estimated loss of 21% of the original global wetland area since 1700 AD[8]. The recent ruling by the U.S. Supreme Court in *Sackett et ux. v. Environmental Protection Agency* et al. potentially enables more wetland loss. Estimating SOC lost from anthropogenic disturbance requires comprehensive SOC mapping that accounts for high SOC in forested wetlands and wet areas which are not contained in contemporary inventories. Omitting these high SOC stocks propagates a potential underestimation of the terrestrial carbon stock in forested regions which contain SOC stores that have accumulated over centuries making them invaluable but irrecoverable if lost within the timeframe to reach net-zero emissions[9].

Freshwater inland wetlands make up 95% of the wetland area in the United States and contain a total SOC stock 8–10-fold higher than the total SOC stock in tidal wetlands[10,11]. Within the inland wetland population, forested wetlands cover the largest extent, but represent the most difficult wetland mapping category to detect, especially in

[1]School of Environmental and Forest Sciences, University of Washington, Seattle, Washington, WA, USA. [2]Department of Forest Resources, University of Minnesota, St Paul, MN, USA. [3]Pacific Northwest Research Station, U.S. Department of Agriculture Forest Service, Juneau, AK, USA. ✉e-mail: ajs0428@uw.edu

satellite and aerial imagery, due to the canopy coverage, small surface area, and isolation from surface waters[12]. Despite the limited appearance, forested wetlands have interconnective roles within terrestrial carbon cycle in addition to SOC storage, including but not limited to: accumulating carbon in aboveground biomass[13]; transporting of labile dissolved organic matter to streams[14]; supplying dissolved $CO_2$ to surface waters leading to significant outgassing[15]; and potentially acting as the highest non-ebullitive $CH_4$ flux from groundwater through tree stems[16]. This diverse array of carbon functions highlights forested wetlands role as a hotspot or ecosystem control point within a landscape[17]. Indeed, seemingly isolated wetlands can connect to surface waters through groundwater links throughout a catchment[18,19] and integrating previously unidentified 'cryptic' forested wetlands can better explain catchment scale surface water chemistry patterns[20]. Cryptic wetlands can also act as the transition between terrestrial and aquatic environments where rapid biogeochemical cycling can occur in spaces only a few meters wide[21]. Mapping SOC along the terrestrial-aquatic gradient containing forested wetlands can reveal hidden SOC spatial patterns that help balance carbon budgets in heterogenous landscapes[22]. However, mapping the distribution of SOC stocks within forested landscapes is challenging, especially with small forested wet areas that do not exhibit conspicuous wetland indicators such as signs of water saturation affecting the aboveground vegetation[23].

Maps of SOC stocks are commonly generated with digital soil mapping (DSM) using geospatial land cover maps and remote sensing metrics relating to the spatial variation of soil forming factors[24]. Wetlands integrated into DSM are often inconsistently defined with insufficiently measured wetland extent that promotes underestimation and inaccurate spatial distributions of SOC stocks[25]. Yet, generally, wetland mapping continues to improve with machine learning models utilizing geospatial wetland and peatland soil properties, but there is still substantial variation and underestimates in forested wetlands and wet areas[26]. In areas where forest canopy obscures wet areas, data driven machine learning approaches utilizing topography focused metrics can identify previously hidden forested wetlands and wet areas by capturing patterns of surface and groundwater flow that facilitate water accumulation within a landscape[27]. Utilizing continuous probabilities simulated from presence/absence data[28], probabilistic wetland mapping can capture the spatial representation of the terrestrial to aquatic gradient, with wetlands as one end of a water saturation continuum[29,30]. SOC is expected to increase with the higher probability of a wetland where soil saturation that inhibits microbial respiration and facilitates organic matter accumulation and potential wetland extent can be estimated above a chosen probability threshold[31,32].

We have yet to note SOC maps informed by potential wetland presence which: (1) identifies unmapped SOC in potential wetland area; (2) compares potential wetland SOC with maps of existing wetland SOC estimates; and (3) compares overall SOC distributions with available SOC mapping products. Here, we conduct a DSM SOC mapping approach in the Hoh River Watershed (HRW), a densely forested, geomorphologically complex watershed using a continuous probabilistic wetland identification metric to reveal significant amounts of unmapped SOC contained in potential forested wetlands and wet areas. We adapt the term 'cryptic wetland' from ref. 20 as 'cryptic carbon' to distinguish hidden SOC stocks within potential forested wetlands that have not been mapped or estimated previously, with the caveat that we are not mapping jurisdictional wetland boundaries. Our approach identifies a nearly five-fold increase in the amount of SOC estimated to be contained in potential wetlands, of which is mostly hidden under thick forest cover. This approach is adaptable and flexible for natural resource managers and conservationists to identify potentially immense cryptic carbon stocks that have not been associated with potential forested wetlands.

## Results

### Field collected pedon SOC stocks

We investigated the distribution of SOC stock across the field collected pedon sample depth profile where the overall mean pedon depth was $95 \pm 4.4$ cm (standard error of the mean $(\frac{\hat{\sigma}^2}{\sqrt{n}})$) with $94 \pm 5.4$ cm for uplands and $99 \pm 6.5$ cm for wetlands. Within wetland pedons, 38% of the entire SOC stock was in 0–30 cm, 31% in 30–60 cm, 27% in the 60–100 cm, and 4% in 100–120 cm (Table 1). Within upland pedons, 49% of the SOC stock was in 0–30 cm, 32% in 30–60 cm, 16% in 60–100 cm, and 3% in the top 120 cm. Overall, 96% and 97% of the entire soil carbon stock was contained in the top 1 m of the soil profile for wetlands and uplands, respectively, which we used as a standardized depth for spatial predictions across the HRW. Mean 1 m depth SOC stocks within our field pedon dataset was $221 \pm 27.0$ MgC ha$^{-1}$ standard error of the mean $(\frac{\hat{\sigma}^2}{\sqrt{n}})$. Wetlands in our field pedon dataset contained a higher mean 1 m SOC stock of $346 \pm 89.1$ MgC ha$^{-1}$ which was also much higher compared to $185 \pm 20.2$ MgC ha$^{-1}$ in uplands (Table 1). Within wetlands, we classified riverine and palustrine wetlands due to differences in soil parent material leading to significant differences in SOC. Palustrine wetlands defined here are similar to the Cowardin classification adapted by the NWI[12] of any freshwater (or less than 0.5 ppt salt concentration), non-tidal, non-riverine, or non-lacustrine wetland, inclusive of forested and non-forested vegetation. Palustrine contained a mean 1 m SOC stock of $447 \pm 81.6$ MgC ha$^{-1}$ compared to a mean of $43.3 \pm 11.7$ MgC ha$^{-1}$ in riverine wetlands. Palustrine wetland SOC stock distribution in the soil profile was 37% in the top 30 cm, 31% in 30–60 cm, and 28% in the 60–100 cm. Riverine wetland SOC stock distribution was mostly contained in the top 30 cm (96%). We noted in field observations that pedon locations with WIP probabilities between 25–50% that appeared to maintain a mesic soil moisture environment between wetland and upland ends of the WIP probability range. Pedons within the mesic zone contained a 1 m mean SOC stock of $241 \pm 36.5$ MgC ha$^{-1}$ SOC stock which is elevated above uplands in our dataset and within the standard error range of the overall WIP wetland class. Pedons with WIP probabilities below 25% contained a mean of $149 \pm 19.8$ MgC ha$^{-1}$.

### Model predictions and mapping of SOC stocks

We used the model shown in Eq. 2 and graphed in Fig. 1 to predict SOC stocks across the HRW (Fig. 2). From these predicted maps, we calculated a mean 1 m SOC stock of $127 \pm 26.0$ (87–178) MgC ha$^{-1}$

**Table 1 | Soil organic carbon (SOC) stocks, sample depths, and sample numbers collected in the Hoh River Watershed (HRW)**

| Landscape Class | 30 cm SOC Stock | 60 cm SOC Stock | 1 m SOC Stock | 120 cm SOC Stock | Sample Depth | n |
|---|---|---|---|---|---|---|
| WIP Wetland | 138 ± 22.9 | 250 ± 53.4 | 346 ± 89.1 | 362 ± 96.9 | 99 ± 6.5 | 8 |
| Riverine Wetland* | 41.7 ± 12 | 42.8 ± 11.6 | 43.3 ± 11.7 | 43.3 ± 11.7 | 81 ± 4.0 | 2 |
| Palustrine Wetland* | 171 ± 11.8 | 320 ± 38.7 | 447 ± 81.6 | 468 ± 92.3 | 110 ± 6.9 | 6 |
| WIP Upland | 93.6 ± 7.65 | 153 ± 15.1 | 185 ± 20.2 | 190 ± 21.2 | 94 ± 5.4 | 28 |
| All Landscapes | 104 ± 8.27 | 175 ± 17.7 | 221 ± 27.0 | 228 ± 28.8 | 95 ± 4.4 | 36 |

The ±indicates standard error. The *indicates landscape class subsets from wetlands determined by the Wetland Intrinsic Potential (WIP) tool.

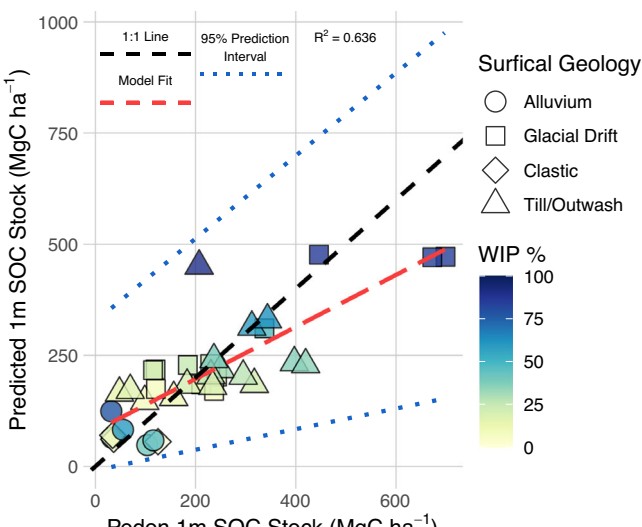

**Fig. 1 | Model predicted soil organic carbon (SOC) stock compared to field sampled pedon SOC stock with a gradient of probability from the Wetland Intrinsic Potential (WIP) tool.** Modeled SOC stocks were back transformed from square root values in the original linear mixed effects model with the fixed effect of WIP probability and random effect of surficial geology. Shading represents WIP probability. Shapes represent different surficial geology categories. Prediction intervals are based on bootstrapped 95% confidence intervals.

(±standard deviation, parentheses contain 95% confidence interval from the bootstrapped mapped predictions) and a mean 30 cm SOC stock of 72.9 ± 12.5 (55.1–103) MgC ha⁻¹ (See Methods for model fit evaluation, Table 2. for tabulated 1 m and 30 cm SOC stocks, Supplementary Fig. 4 for mapped 1 m SOC stock standard deviation and Supplementary Figs. 5 and 6 for mapped mean and standard deviation of 30 cm SOC stocks and Supplementary Fig. 7 for 30 cm model prediction vs. actual scatterplot). The overall total 1 m and 30 cm SOC stocks of the HRW were 8.6 ± 1.8 (5.9–12.1) TgC and 5.0 ± 0.9 (3.8–7.0) TgC, respectively. We focus on the 1 m SOC stocks for mapped predictions with wetlands. Comparisons where wetlands defined by the WIP probability ≥50% covered 6115 ha of the HRW and contained a mean 1 m SOC stock of 277 ± 49.7 (197–383) MgC ha⁻¹ which was more than twice as high as the overall HRW concentration and the mean upland SOC concentration of 112 ± 23.6 (75.7–157) MgC ha⁻¹. Wetlands in our study also contained disproportionately more SOC for a total of 1.7 ± 0.3 (1.2–2.3) TgC SOC stock or 20% of the overall HRW SOC stock in 9.0% of the total landscape surface area for an SOC:Extent ratio of 2.2. Comparatively, uplands contained 6.9 ± 1.5 (4.7–9.8) TgC or 80% of the HRW SOC stock in 91% of the HRW surface area for a SOC:Extent ratio of 0.9. Within overall wetlands, we identified 4935 ha of forested wetlands with canopy coverage ≥50% that contained higher mean SOC stocks of 292 ± 50.5 (210–399) MgC ha⁻¹ for a total of 1.4 ± 0.2 (1.0–2.0) TgC SOC stock. These forested wetlands composed 81% of the overall wetland extent and 85% of the overall WIP wetland SOC stock for an SOC:Extent ratio of 2.3. Of the two wetland types delineated by surficial geology, riverine wetlands covered 1726 ha and lower mean SOC stocks of 101 ± 34.7 (50.4–181) MgC ha⁻¹ and a total SOC stock of 0.2 ± 0.1 (0.1–0.3) TgC and 0.8 SOC:Extent ratio. Conversely, palustrine wetlands contained a significantly higher 347 ± 55.7 (255–463) MgC ha⁻¹ SOC stock and totaled 1.5 ± 0.2 (1.1–2.0) TgC or 18% of the total landscape SOC within 6% of the surface area of the HRW for a 2.7 SOC:Extent ratio.

Wetland SOC stocks measured from the National Wetland Condition Assessment (NWCA) and U.S. Department of Agriculture's National Cooperative Soil Survey (NCSS) Soil Survey Geographic Database (SSURGO) datasets were upscaled with wetland extent from the National Land Cover Database (NLCD) in ref. 11 and termed NWCA-SSURGO for reference. Wetlands in NWCA-SSURGO contained a mean stock of 184 ± 108 MgC ha⁻¹ (standard deviation only, ref. 11 did not report 95% confidence intervals; Fig. 3 inset and Fig. 4c). Using the 1640 ha wetland extent measured within the HRW for the NWCA-SSURGO dataset, we calculated a total of 0.3 ± 0.2 TgC across the HRW for a SOC:Extent ratio of 1.4 (Table 2). Within the total NWCA-SSURGO wetlands, forested wetlands, defined by the canopy coverage ≥50%, comprised 90% of the wetland SOC and 88% of the wetland extent. Compared to our WIP-derived wetland SOC estimates, the mean concentration NWCA-SSURGO wetland SOC stock was approximately two-thirds or 66% of the mean WIP wetland SOC stock. Due to the large differences in wetland extent, the total wetland SOC of the WIP-derived estimates (1.7 TgC) was 462% higher than the total wetland SOC stock in NWCA-SSURGO (0.3 TgC) showing that only 18% of the total potential wetland SOC stock is currently mapped. By removing overlapping wetland areas covered by the NWCA-SSURGO datasets within our WIP dataset we estimated 5308 ha of unmapped potential wetlands which we designate as cryptic carbon. This cryptic carbon contained a mean SOC stock of 275 ± 49.2 (196–379) MgC ha⁻¹ and a total SOC stock of 1.5 ± 0.3 (1.0–2.0) TgC for an SOC:Extent ratio of 2.2 (Table 2). The total SOC stock of cryptic carbon is 382% higher than the currently mapped total wetland SOC in NWCA-SSURGO estimates and approximately 17% of the total HRW SOC stock from our model. Within cryptic carbon, 80% is considered forested with canopy cover ≥50% and contains 84% of the total cryptic carbon SOC. Adding the total cryptic carbon SOC stock of 1.5 ± 0.3 TgC to the 0.3 ± 0.2 TgC in the NWCA-SSURGO increases total wetland SOC stock in the HRW by 482% to 1.8 ± 0.2 TgC (note, 95% confidence intervals removed from combined total, see Supplementary Table 2 for details) and more than quadruples the estimated SOC stored in wetlands.

In our analysis of the wetland extent distribution from the WIP model (Fig. 3b), our minimum wetland extent ranged from 64 m² or 0.0064 ha to the largest wetland with 400 ha. In total, we found 31,981 individual wetlands of which approximately 96% were smaller than the minimum mapping unit of 1 acre (0.40 ha) used by the NWI (Supplementary Table 3). After extracting SOC stocks from our earlier WIP-based model prediction, the SOC distribution across WIP wetlands sizes showed that a majority of wetland surface area (86%) and SOC stock (87%) was contained in wetlands greater than 1 acre (0.40 ha). Indeed, the extent of each of the largest 5 wetlands were all greater than 100 ha, the largest of which was a 400 ha wetland containing 0.15 TgC or 18% of the total wetland SOC stock (1.7 TgC). The relationship between SOC stock and individual wetland extent was shown to be linear in a log-log plot indicating that there is a non-linear increase in total SOC stock with increasing wetland extent (Supplementary Fig. 8). Mean stock SOC density across the size distribution was consistent around 250 ± 40.8 (185–335) MgC ha⁻¹ to 252 ± 42.0 (185–340) MgC ha⁻¹ with slight increase with surface area with the smaller wetlands containing 251 ± 40.5 (187–336) Mg ha⁻¹ compared to the largest wetlands containing 264 ± 47.8 (188–367) Mg ha⁻¹ (Supplementary Table 3).

## Discussion

Our results show continuous representation of potential wetlands and that wet areas integrated into DSM SOC mapping approach greatly improves the spatial representation of SOC. The results explicitly show high SOC stocks in potential wetland areas along with gradients between wetland and upland areas corresponding to the terrestrial to aquatic gradient. Overall, the spatially continuous WIP probability metric was a significant covariate for SOC when combined with surficial geology corresponding to soil parent material and enabled wall-to-wall mapping across the large heterogenous and geomorphologically complex HRW catchment. Probabilistic modeling of wetland presence has become increasingly relevant in wetland mapping research instead

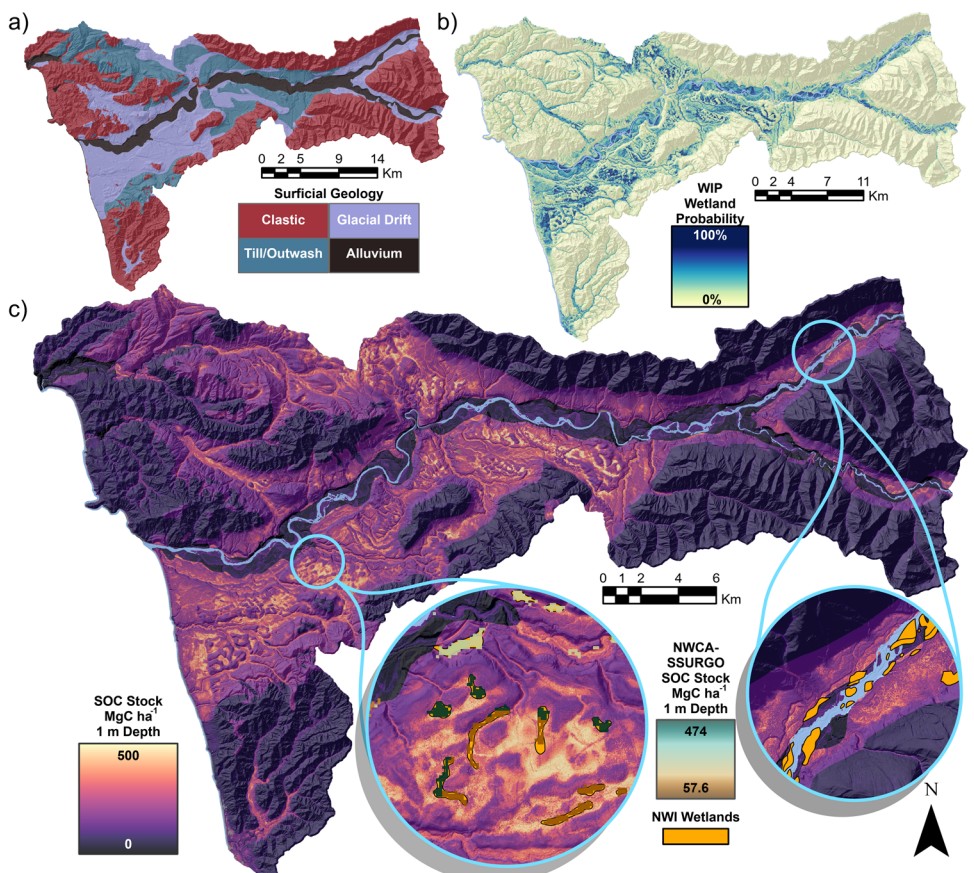

**Fig. 2 | Maps of surficial geology and Wetland Intrinsic Potential (WIP) probability parameters and soil organic carbon (SOC) model output in the Hoh River Watershed (HRW). a** Shows the surficial geology categories of the HRW by color classes in surficial geology legend, **b** shows the WIP probability gradient shown by yellow-blue shading indicated in WIP legend, **c** shows the predicted 1 m SOC stock across the HRW with purple-to-yellow shading that continues in inset maps showing fine scale SOC patterns overlain by estimated SOC shown by brown-teal shading from the harmonized National Wetland Condition Assessment and Soil Survey Geographic Database (NWCA-SSURGO) dataset in ref. 11 and additional current wetland extent from the National Wetland Inventory (NWI). We added a semi-transparent hill shade layer to highlight terrain and removed the river surface water shown in light blue for the final prediction map.

of more discrete land cover classification[29] and the WIP probability model generated with fine resolution topography metrics, identified cryptic wetland features beneath a forest canopy[33].

The statistical model we developed to predict SOC stocks from the WIP probability is simple and conservative, appearing to underestimate large wetland SOC stocks shown by the underprediction of the upper range of SOC values in Fig. 1. The treatment of surficial geology as a random effect which shrinks overall estimates towards an overall mean driven by the WIP. Finer scale surficial geology and other drivers of SOC, such as primary productivity are present but not accounted for in the model, but we explain substantial variation with our prediction ($R^2 = 0.63$). We found that cryptic carbon defined as the WIP-defined wetland SOC outside of currently mapped areas contains the majority of the modeled wetland SOC stock with approximately 86% of the wetland SOC. Consequently, cryptic carbon added to current NWCA-SSURGO estimates increased the total wetland SOC stock in the HRW from 0.3 TgC to 1.8 TgC or by 482%. Most of the cryptic carbon stock is due to the 273% increase in potential wetland extent from the WIP ≥ 50%. However, identification of previously omitted wetland extent using the WIP tool contained wetlands with a higher mean SOC stock of $259 \pm 72$ (187–331) MgC ha$^{-1}$ compared to the mean SOC stock in NWCA-SSURGO wetlands ($184 \pm 108$ MgC ha$^{-1}$) showing a new inclusion of wetlands with high SOC stocks. The wetland SOC measured in this study was predominantly contained in the first 1 m of soil depth and within large wetland extents although there are potentially numerous small wetlands within the HRW. We note,

however, that wetlands identified within our study do not represent jurisdictional wetlands or delineate wetland boundaries, nor do they represent the greater population of wetlands outside our study area. But the framework of our study begins to address the critical gap in omitting wetlands and wet areas in SOC mapping showing significant SOC underestimates when upscaling current wetland SOC data (NWCA) with optical imagery based landcover datasets (NLCD). This study provides an initial step toward improving wetland carbon monitoring systems.

While our study focused on potential wetland SOC, upland SOC is the largest fraction of the total HRW SOC stock. Two global models that provide readily accessible gridded 30 cm SOC maps are SoilGrids 2.0[6] and the Global Soil Organic Carbon (GSOC)[34] Map. Our estimates of mean 30 cm SOC stock are lower than those of SoilGrids 2.0 but higher than GSOC (Supplementary Table 4) indicating appropriately estimated SOC stock magnitude at the lower end of the WIP probability which may represent soil moisture regimes and their control on SOC in non-wetland areas. Compared to GSOC and our results, SoilGrids 2.0 potentially overestimates SOC stocks in the HRW but is also within the range of other studies using data from the National Forest Inventory[35]. Soils are typically carbon dense in the Pacific Northwest region due to the humid temperate climate of the region and tends to be higher than SOC measurements in other systems for both wetlands and uplands[36]. The region our study takes place in, is the southern portion of the North Pacific Coastal Temperate Rainforest, a region that expands north to central Alaska. For this same region ref. 37

**Table 2 | Metrics from mapping 1 m and 30 cm depth soil organic carbon (SOC) stocks across the Hoh River Watershed (HRW)**

| Source | Landscape Class | Surface Area (ha) | 30 cm Depth Mean SOC Stock (MgC ha⁻¹) | 30 cm Depth Total SOC Stock (TgC) | 1 m Depth Mean SOC Stock (MgC ha⁻¹) | 1 m Depth Total SOC Stock (TgC) |
|---|---|---|---|---|---|---|
| WIP | Wetland | 6115 | 122 ± 14.0 (101–153) | 0.7 ± 0.1 (0.6–0.9) | 277 ± 49.7 (197–383) | 1.7 ± 0.3 (1.2–2.3) |
| | Riverine* | 1726 | 59.6 ± 18.4 (35.1–101) | 0.1 ± 0.0 (0.1–0.2) | 101 ± 34.7 (50.4–181) | 0.2 ± 0.1 (0.1–0.3) |
| | Palustrine* | 4390 | 146 ± 12.2 (126–174) | 0.6 ± 0.1 (0.6–0.8) | 347 ± 55.7 (255–463) | 1.5 ± 0.2 (1.1–2.0) |
| | Forested* | 4935 | 127 ± 13.4 (106–157) | 0.6 ± 0.1 (0.5–0.8) | 292 ± 50.5 (210–399) | 1.4 ± 0.2 (1.0–2.0) |
| | Upland | 62,030 | 68.1 ± 12.4 (50.6–97.9) | 4.2 ± 0.8 (3.1–6.1) | 112 ± 23.6 (75.7–157) | 6.9 ± 1.5 (4.7–9.8) |
| | Total HRW | 68,145 | 72.9 ± 12.5 (55.1–103) | 5.0 ± 0.9 (3.8–7) | 127 ± 26.0 (86.6–178) | 8.6 ± 1.8 (5.9–12) |
| NWCA-SSURGO | Wetland | 1640 | 89.2 ± 51.8 | 0.1 ± 0.08 | 184 ± 108 | 0.3 ± 0.2 |
| | Forested* | 1442 | 90.6 ± 51.8 | 0.1 ± 0.08 | 188 ± 110 | 0.3 ± 0.2 |
| WIP – (NWCA-SSURGO) | Wetland | 5308 | 121 ± 13.9 (99.9–152) | 0.6 ± 0.1 (0.5–0.8) | 275 ± 49.2 (196–379) | 1.5 ± 0.3 (1.0–2.0) |
| | Forested* | 4236 | 127 ± 13.3 (106–157) | 0.5 ± 0.1 (0.4–0.7) | 290 ± 50.0 (209–396) | 1.2 ± 0.2 (0.9–1.7) |
| Combined: WIP + (NWCA-SSURGO) | Wetland | 6948 | 114 ± 22.9 | 0.8 ± 0.2 | 253 ± 63.0 | 1.8 ± 0.4 |
| | Forested* | 5678 | 117 ± 23.3 | 0.7 ± 0.1 | 264 ± 65.3 | 1.5 ± 0.4 |

The ± indicates the standard deviation of the bootstrapped model predictions for the Wetland Intrinsic Potential (WIP) derived estimates and published standard deviations from the harmonized National Wetland Condition Assessment and Soil Survey Geographic Database dataset (NWCA-SSURGO) datasets in ref. [11] Landscape class metrics were determined by masking the map of 1 m SOC stocks with surficial geology, canopy cover ≥50%, or WIP ≥50%. The * indicates the landscape class subsets from overall wetlands in both WIP & NWCA-SSURGO maps. We only show ± standard deviations for combined metrics.

measured a median SOC stock of 168.4 MgC ha⁻¹ and mapped mean SOC stock of 228 ± 111 MgC ha⁻¹, much higher than our overall mapped SOC of 127 ± 26 (87–178) MgC ha⁻¹ in the HRW. This discrepancy is likely due to the presence of numerous northern peatland SOC stocks >500 MgC ha⁻¹ in the region. Peat formation is more frequent farther north in cooler and wetter climates and can accumulate organic material in deposits as deep as 3–5 m[26]. Our estimates in the HRW are potentially missing these high SOC stocks due to lack of peat samples in the sampling scheme and limiting the model prediction to 1 m depth. Similarities to the remote-sensing driven peatland probability model developed by ref. [29] show our approach could apply towards landscape areas containing peatlands but additional classification may be needed since peatlands store significantly more SOC than mineral soil wetlands per unit area[38].

For wetland-specific SOC stocks across different climatic zones in CONUS, ref. [11] measured 114.8–398.5 MgC ha⁻¹ wetland SOC stocks with higher SOC stocks in the Eastern Mountain region due to presence of peatlands and lower SOC stocks in the arid West and Coastal Plains. Lower wetland SOC stocks are prevalent in more arid regions shown by Tangen and Bansal[39] who measured 81.97 MgC ha⁻¹ wetland SOC in the semi-arid prairie pothole region. The forested wetlands in our study can be compared to findings from ref. [13] who measured mean forested wetland SOC concentrations across the Eastern-to-Midwest U.S. and Canada ranging from 165 ± 12 MgC ha⁻¹ to 264 ± 46 MgC ha⁻¹ noting the highest amounts in broad-leaved and shrub/thicket wetland types and lowest in needle-leaved forests. However, it is not uncommon for SOC to be higher in needle-leaved forests, which can accumulate significant amounts of carbon in colder and wetter climates[40].

Although SOC stocks vary between wetlands, SOC stocks can also vary within individual wetlands[41]. Tangen and Bansal[39] showed significant differences in SOC stocks between different landscape positions within individual wetlands and landscape position factored heavily into restored wetland SOC stock. Stewart et al.[42] showed that although there was considerable variability of SOC stocks within wetlands, terrain metrics related to hydrology explained a significant amount of the variation, and further pointed towards topography enhancing large-scale analyses. This inference on topography informing larger scale analysis corresponded well to our use of the WIP tool in this current study, which relies on topographic metrics calculated at different scales[33]. Because of the continuous gradient produced by the WIP, we speculate that some intra-wetland variability may be accounted for in the SOC stock map. However, more explicit intra-wetland sampling would be necessary to support this notion.

Applying the framework of SOC modeling based on wetland probability in and across larger regional and continental scales would require adjusting the wetland probability with additional covariates corresponding to climatic controls on SOC in order to accurately represent changes in SOC accumulation in different wetland types in different locations. The HRW as a single watershed does not represent most of the forested watersheds in CONUS, which inhibits extrapolating SOC stock numbers across larger extents, particularly in non-temperate regions. However, we have shown that the large increase in wetland extent corresponds to substantial shift in the landscape spatial pattern of SOC stocks. Moreover, these results support a critical need to evaluate the wetland SOC stocks in forested regions, particularly the large central and eastern temperate hardwood forests, which contained large wetland SOC stocks shown in NWCA data from Nahlik and Fennessey[10] and NWCA-SSURGO data from ref. [11]. Additional improvements to mapping SOC would also include modeling the probability of different wetland classes, especially peat-forming wetlands, by using classification data available from open sources such as the NWI and NLCD[43].

The wetlands we identified with the WIP and not included in the NLCD maps used to upscale the NWCA-SSURGO wetland SOC estimates contained 1.5 TgC or 86% of the 1.7 TgC total HRW wetland SOC,

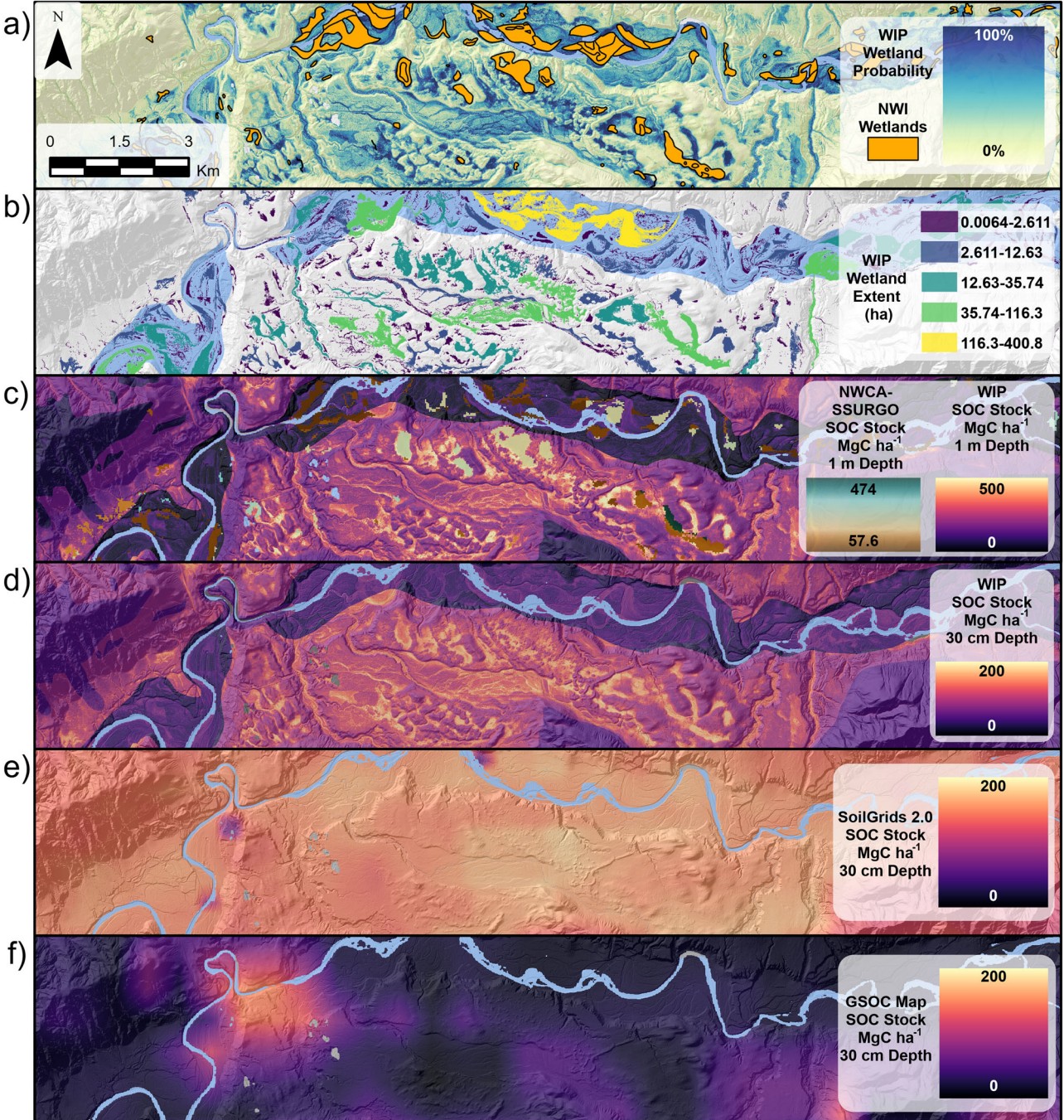

**Fig. 3 | Maps of wetland surface areas and soil organic carbon (SOC) stocks compared current national and global map products. a** Shows the Wetland Intrinsic Potential (WIP) tool wetland probability gradient shown by yellow-blue shading overlain by orange National Wetland Inventory (NWI) wetlands; **b** shows size classes of wetland extent colored by size ranges where wetlands are defined WIP ≥ 50%; **c** shows the 1 m WIP modeled SOC stock distribution with purple-to-yellow shading overlain by SOC estimates from National Wetland Condition Assessment and Soil Survey Geographic Database (NWCA-SSURGO) dataset shown by brown-teal shading; **d** shows the 30 cm WIP modeled SOC stock distribution with purple-to-yellow shading; **e** shows the 30 cm SoilGrids 2.0[6] modeled SOC stock distribution with purple-to-yellow shading; and **f** shows the 30 cm SOC stocks from the United Nations Forest and Agriculture Organization (UN FAO) Global Soil Organic Carbon (GSOC) Map[34] with purple-to-yellow shading. All maps have an added a semi-transparent hill shade layer to highlight terrain and removed the river surface water shown in light blue.

constituting a substantial cryptic carbon stock. The HRW wetlands estimated by the WIP are dominated by forested wetlands, which we estimate conservatively with tree cover >50% compared to 30% tree cover in the Cowardin classification system[12]. Although the climate, forest species composition, and landform distribution in the HRW are not fully representative of forested regions in the rest of CONUS, we believe cryptic carbon can be a significant proportion of the forested

wetland inventory of CONUS. Forested wetlands are the most extensive wetland type in CONUS according to the inventory of freshwater forested wetlands in the NLCD[11] and in the NWI[12] and approaching the estimation of SOC stocks across CONUS would require a more extensive analysis with additional bioclimatic, physical, and anthropogenic parameters. For example, the Eastern CONUS features extensive deciduous forests with contrasting leaf phenology compared to the

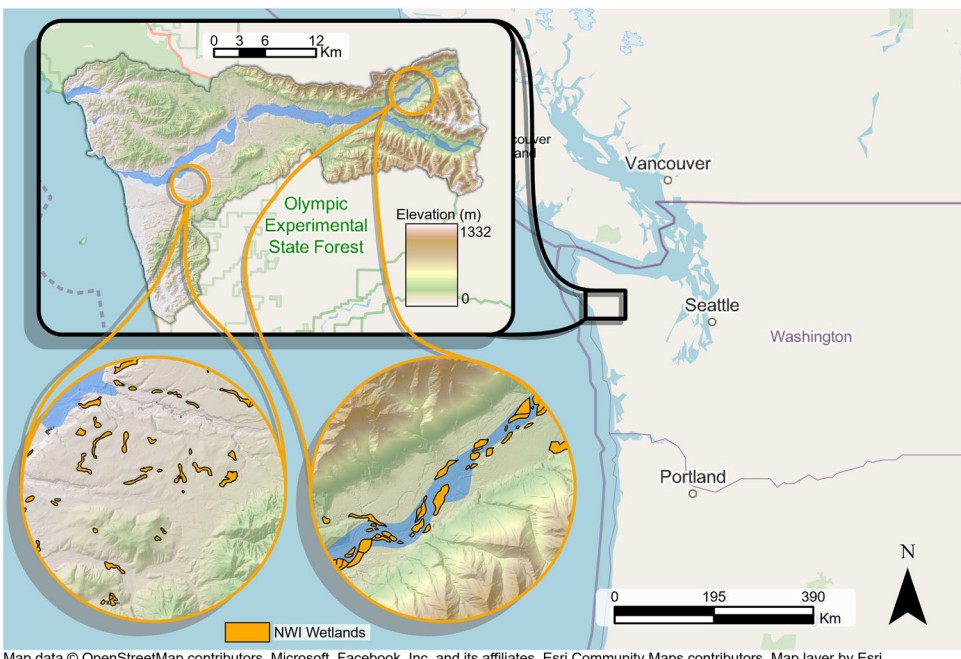

**Fig. 4 | The region of the United States with the location of the Hoh River Watershed (HRW) with National Wetland Inventory (NWI).** The HRW is located in the Pacific Northwest of the United States on the coast of Washington State. The eastern portion of the HRW is mountainous and drains westward to the ocean, which is highlighted by the shaded elevation and topography. Circle inset maps show the eastern and western portions of the lower watershed with wetlands from the NWI colored in orange. We color the river basin area with light blue for the HRW. Base map image credits are listed below the map figure. Regional base map imagery provided by OpenStreetMap under the Open Data Commons Open Database License (ODbL) v1.0 (https://opendatacommons.org/licenses/odbl/1-0/).

coniferous forests of the HRW, which contributes to differences in forested wetland SOC stocks for each forest type[13]. Further, other CONUS regions may have more subtle landforms and topography compared to the HRW, which can affect the rate of forested wetland carbon sequestration[44]. Notwithstanding, many forested wetlands remain mis-classified as upland forest due to difficulty in confirming soil moisture conditions due to dense canopies that obscure surface features from remote imagery. We suggest that there is significant potential to improve on the wetland SOC upscaling by replacing the NLCD wetland mapping from ref. 11 which leveraged the publicly available SOC data from NWCA and Nahlik and Fennessey[10]. The NWCA dataset in Nahlik and Fennessey[10] was sampled to represent $38.4 \times 10^6$ ha total wetlands in CONUS measured as different wetland types by the U.S. Fish and Wildlife Service Status and Trends of Wetlands in the NWI program[12]. Forested wetlands in Nahlik and Fennessey[10] NWCA SOC data contained mean concentrations of $283 \pm 35.4$ MgC ha$^{-1}$ and extrapolation to the $20.1 \times 10^6$ ha in forested wetlands of CONUS inventory resulted in a total forested wetland SOC stock of 5.92 PgC. Uhran et al.[11] updated wetland SOC estimates from Nahlik and Fennessey[10] by revising the NWCA SOC dataset, harmonizing it with the hydric soil data in the Soil Survey Geographic Database (SSURGO), and extrapolated using wetland extent maps from the NLCD. Uhran et al.[11] reported $191 \pm 103$ MgC ha$^{-1}$ mean SOC stock density over $33.9 \times 10^6$ ha of woody wetland extent for a total stock of 6.49 PgC in woody wetlands. We emphasize any extrapolation is highly uncertain but using our example of missing extent in the HRW, a more than 3-fold increase in wetland extent (1640 ha in NWCA-SSURGO to 6949 ha in WIP + NWCA-SSURGO) could substantially drive a similar increase in cryptic carbon within forested areas of CONUS which we estimate to be up to $20 \pm 13$ PgC or $31 \pm 11\%$ (standard deviation) of the current 65 PgC total CONUS 1 m SOC stock[45,46] calculated by extrapolating the mean and uncertainty from Uhran et al.[11] and Nahlik and Fennessey[10] by the 324% increase in wetland extent observed in this study (6949 ha in WIP + NWCA-SSURGO area compared to 1640 ha in NWCA-SSURGO only). The underestimation of wetland SOC stocks due

to wetland omission has been proposed in previous research using currently available large scale maps of land cover[47] and is clearly shown here using products like the NLCD. Publicly available datasets such as the NWI, NWCA, and SSURGO will continue to add data and improve in the future and represent opportunities to improve upscaling wetland SOC estimates at larger regional scales with more comprehensive and accurate wetland extent maps. Our analysis supports improving SOC distribution estimates with new approaches using topography-based probabilistic models that better reflect the spatial distribution of soil moisture conditions and provide a continuous, spatially-explicit upscaling metric.

Due to the forested overstory, cryptic carbon is likely to experience deforestation as a disturbance but its detection and frequency is unknown due to its omission from current wetland maps and inventories[12]. The SOC stored as cryptic carbon depends on consistently wet soil conditions and forest harvest practices negatively affect SOC stocks by utilizing intensive site preparation through draining wet areas for tree extraction[48]. Cryptic carbon in headwaters may be especially sensitive to hydrologic disturbance from forestry activities due to more intimate connections to groundwater[49]. Removal of forest canopy in forested wetlands and exposing soil to warmer temperatures can lead to higher rates of SOC decomposition[50]. But long periods of recovery post-harvesting can allow SOC and soil nutrients to return to pre-harvest levels ameliorating impacts on forest wetland function[51]. The effects of deforestation on forested wetlands will also vary by ecosystem type and region. Significant SOC stock destabilization and export of fluvial organic carbon was found in tropical forested wetlands and peatlands that experience deforestation and drainage[52]. More work is needed to improve wetland mapping under forest canopy in tropical regions, which are one of the largest sources of uncertainty in the global carbon cycle[53]. While deforestation itself may not lead to complete wetland drainage, land use conversion to agriculture is another persistent threat to wetlands that more effectively drains wetlands and produces substantial carbon release as greenhouse gases[54].

Globally since 1700, the main driver of wetland loss has been drainage and conversion to agriculture with regional hotspots in the United States, Europe, Central and Southeast Asia, and Japan[8]. Land use conversion is a top contributor to carbon emissions after fossil fuels and is driven mostly by deforestation[55]. Carbon emissions and losses of SOC from disturbed soils are more uncertain than forest biomass carbon loss in the global carbon budget[56] but as much as 133 PgC of SOC has been lost from soils over the course of 12,000 years of human agriculture[57]. There is yet to be a consensus estimate of total global SOC losses due to wetland conversion, but there is consistent evidence of increased carbon emissions and SOC loss when wetlands, and in particular peatlands, are drained and converted to agriculture[58–61]. Utilizing the newest global wetland maps, which model inundation frequency could help improve spatial estimates of SOC with large global soil pedon databases[4,6]. At the continental or national scale, research with moderate resolution remote sensing from Landsat has been used to map wetland extent with SOC stock declines showing significant reductions in the last half century[62]. Similar approaches can be applied at the large catchment scale, such as our current study in the HRW which could potentially provide contemporary insight to wetland SOC changes since the lidar acquisition in 2012 and 2013. The wetland types examined here and in other regions also experience SOC stock destabilization and emit previously stored carbon as $CO_2$ due to conversion to cropland[10,39]. It is uncertain how the inclusion of cryptic carbon stocks will affect the total estimates of wetland SOC stock affected by disturbance and the magnitude of potential SOC release as $CO_2$. But more accurate mapping of forested wetland extent and SOC stock will improve conservation of a valuable carbon sink that is underestimated with currently mapped wetland extents.

Our study provides an adaptable approach that is informed by a continuous wetland identification metric which maps and reveals high SOC stocks driven by wetland potential on the landscape. This mapping revealed the vast stores of unmapped forested wetland SOC stocks or cryptic carbon compared to currently available wetland SOC maps. We show cryptic carbon contained a higher mean SOC stock than both currently mapped wetlands and uplands. When added to currently available estimates of wetland SOC stock in the HRW, cryptic carbon increased the total SOC stock from 0.3 TgC to 1.8 TgC or by 482%. The majority of this cryptic carbon was contained in wetlands greater than 1-acre or 0.4 ha, a common minimum mapping unit. There are still considerable uncertainties in extrapolating SOC increase results to the greater population of forested wetlands in the U.S., but the potential magnitude of cryptic carbon supports the need for more wetland identification in forested regions in ways that can inform SOC spatial patterns. We provide one approach which integrates potential wetlands into a SOC prediction model, but future research should explore variations of this type of modeling. Metrics that represent the landscape as a gradient of wetlands to uplands can better represent the terrestrial to aquatic gradient that includes potential wetlands and, therefore, areas of SOC accumulation. Land and natural resource managers will be able to use this framework to improve future estimates of SOC spatial patterns as well as wetland SOC vulnerability to land use change.

## Methods
### Study area
This study takes place in the Hoh River Watershed (HRW) within the Pacific Northwest of the Conterminous United States (Fig. 4), which contains some of the highest aboveground carbon and SOC stocks in the world reaching 375 MgC ha$^{-1}$ and 709 MgC ha$^{-1}$, respectively[63]. In the HRW, mean annual air temperature is 7.2 °C and mean annual precipitation is 274 cm and can exceed 300 cm with most of the precipitation in winter which mainly falls as rain but snowfall is more common in the upper elevations[64]. The mountains of the HRW were created 17–20 million years ago during the Miocene to Eocene periods

with the uplifting of marine sedimentary rock over the denser ocean crust. The uplifted marine sedimentary rock also formed hills and terraces in the lower HRW. During the end of the Pleistocene and the period of deglaciation, large floods from glacial melt deposited material over the lower elevation so the HRW creating large flood-plains. Rivers continued to incise this deposited glacial material over the Holocene and into the present depositing alluvium near the present main channel of the Hoh river that bisects the HRW[65]. Current topography varies from mountains with steep slopes (>40%) in the eastern portion of the HRW to rolling hills and flat areas in the lower floodplain that drains eastward to the Pacific Ocean. Soils of the HRW reflect this geologic history and topography with dominant soils containing loamy to sandy-clay coarse textures although there is a moderate presence of volcanic ash and which promotes Andisol soil development[66]. The HRW has a mix of both private and public forestlands dominated by Sitka Spruce (*Picea sitchensis*) and Western Hemlock (*Tsuga heterophylla*) in the lower elevations that is actively managed for timber harvest although areas along the coast and in the upper watershed are part of the Olympic National Park with protected old-growth forest containing trees up to 4 m in diameter and 80 m in height[67]. The mapped wetlands within the HRW are diverse, from precipitation-driven bogs to riparian wetlands (Fig. 4 insets).

Many of the wetlands are under dense forest overstory but in some forested areas with high levels of inundation trees are stunted in size and have a lower overall height and biomass. The most prominent Hydrogeomorphic wetland classes are Riverine, Mineral Flats, Organic Flats, and Depressional[68,69]. There is a notable difference between Riverine wetlands and the other wetland classes for SOC and we mark this distinction with grouping all wetland hydrogeomorphic classes into two classes for our soil pedon dataset: Riverine and Palustrine (non-riverine). Palustrine wetlands are similar to the Cowardin classification as any freshwater (or less than 0.5 ppt salt concentration), non-tidal non-riverine, or non-lacustrine wetland, inclusive of forested and non-forested vegetation[12].

### Wetland maps with the Wetland Intrinsic Potential Tool (WIP)
We mapped wetlands using the Wetland Intrinsic Potential (WIP) tool, a multi-scale terrain-based wetland identification and mapping tool developed by ref. 33. The WIP tool models wetland presence in a spatially explicit, continuous pixel approach using input parameters related to hydrophytic vegetation, hydrology, and hydric soils. The topographic and terrain input data layers are derived from discrete point aerial lidar which was processed to create a digital elevation model at a 4 m cell size resolution of the terrain surface (Lidar source: 2012–2013 Puget Sound LiDAR Consortium (PSLC) Topographic LiDAR: Hoh River Watershed, Washington (Deliveries 1 and 2), vertical absolute accuracy RMSE: 0.043 m; vertical relative accuracy RMSE: 0.082 m). Unlike aerial or satellite imagery, lidar can detect small topographic features under tree canopy and terrain metrics were integrated into a random forest model that was trained on wetland presence/absence point datasets derived from the National Wetland Inventory (NWI) and validated with additional field collected ground-truthed datasets. The WIP tool was specifically developed to identify wet areas missing in most wetland inventories because they do not have standing water or are hidden under tree or vegetation canopy making them difficult to detect in satellite or aerial imagery. We refer the readers to Supplementary Methods and ref. 33 for the full summary of how the WIP tool was implemented in the HRW. The output produces a wetland probability score based on the proportion of classification trees in the random forest model of how likely a pixel is a wetland (0–100%), which is the estimated likelihood that the wetland class label is correct for a given input of terrain, hydrology, and vegetation parameters. For example, a pixel that has a wetland probability of 80% will contain a combination of landscape features that generate a wetland within 80% of the dataset. Wetlands, therefore,

represent the high end of continuum corresponding to landscape soil moisture and inundation and the other end is an absence of these conditions. Because the wetland probability is continuous across the entire landscape, it enables SOC stock to be modeled continuously across the entire HRW. However, setting a threshold probability also allows estimates of wetland extent. In order to determine potential wetland extent, we chose the threshold value of 50%, above which classifies a pixel as a wetland and below which classifies pixels as non-wetland or upland. WIP model accuracy for the HRW in Washington State using wetland probability ≥50% to create a binary class of wetland (>50%) vs. upland (<50%) was 93.0%. Readers should consider that wetlands defined by the WIP tool do not have jurisdictional boundaries which require field delineation and verification to determine their exact extent based on hydrology, hydric soil, and hydrophytic vegetation at a much smaller scale.

## Field sampling for soil pedons

We developed a stratified random sampling approach across the HRW WIP probability distribution. Our strategy was to sample at a consistent interval in the distribution to evenly obtain samples and address potential areas of uncertainty. We divided the WIP distribution into 30 probability bins and sampled 1 pedon at a random location per bin then added 6 additional pedons split between the highest and lowest probabilities as time allowed. Once sampling locations were selected, we used Garmin Handheld Global Positioning System (GPS) to navigate to each point. After designating the pedon sampling location, we then used a JAVAD GNSS Triumph-2 for more precise georeferencing. While we strove to remain unbiased in our selection of sample sites, we faced difficulty accessing the precise location of randomly selected, 4 m resolution pixels due to limited precision in GPS navigation equipment which was approximately 4–10 m in ideal open sky conditions and degraded further under canopy. Therefore, the final distribution of WIP probability values for our sampled pedons was not evenly split between wetlands and uplands. We note that uplands compose the majority of the HRW as shown by the histogram of the entire HRW WIP (Supplementary Fig. 3a). Our sampling results somewhat reflect this overall distribution (Supplementary Fig. 3b) and help prevent wetland-bias in our model. In total, we sampled pedons in 8 wetlands and 28 uplands according to the WIP probability ≥50% cutoff for the wetland class from the mapped model. Within the wetland class defined by the WIP ≥ 50%, we classified two distinctive wetland types: riverine ($n = 2$) and palustrine ($n = 6$), which differed in their parent material and organic matter content. Riverine wetlands consisted of recently deposited alluvial material and exhibited very little soil development. We classified these observations in the field and later used a surficial geology map to delineate riverine areas with lower predicted SOC described below.

At each pedon site, a pit was excavated to at least 100 cm depth or to a restricting layer to characterize soil horizons, color, texture, structure, and redoximorphic features[70]. Samples were collected by each soil horizon for bulk density and total carbon analysis. Bulk density was carefully extracted from the pedon face for each horizon using a fixed volume metal cylinder with a volume of 98.175 cm³ for mineral soils or a beveled polyvinylchloride (PVC) cylinder with a volume of 132.536 cm³ for organic soils. Bulk soil samples were taken from each horizon for total carbon analysis. All samples were transported in coolers and stored in refrigerated spaces between 4–6 °C until laboratory preparation and analysis. Laboratory sample preparation included drying all soil samples for at least 48 h or to a constant weight in drying ovens at 75 °C. Soil samples were then sieved to extract the fraction less than 2 mm and remove coarse fragments. Bulk density was calculated as the mass of the less than 2 mm fraction divided by the volume of the fixed volume soil core sampler. SOC was also measured with the <2 mm fraction. Samples were prepared by ball milling a subsample for 2 min at 1/30 second frequency. Then a 20 mg

subsample was run on a Perkin Elmer Co. 2400 model Total Carbon, Hydrogen, and Nitrogen (CHN) Analyzer. SOC stocks for each horizon were calculated from the total carbon percentage from the CHN analyzer (C) multiplied by the bulk density (BD) and the soil horizon thickness (D) (Eq. 1).

$$SOC\ Stock = \sum C_i {}^* BD_i {}^* D_i {}^* (1 - CF_i) \tag{1}$$

Where, $C$ denotes the carbon percentage, $BD$ represents bulk density, $D$ (g cm⁻³) represents the horizon thickness (cm), and $CF$ represents the coarse fragment fraction of the soil sample $i$. For the purpose of this analysis, we do not spatially predict SOC deeper than 1 m soil depth although we collected data beyond 1 m. Soil pedon landscape classes were defined as wetlands for pedons with WIP ≥ 50%, as uplands for WIP < 50%, and as riverine wetland or palustrine wetlands when the sample location was inside or outside the Hoh River floodplain defined by the surficial geology, respectively.

## SOC stock modeling and covariates

All statistical analyses were conducted with R software (version 4.3.0) with dplyr for data management. To generate a prediction model for SOC, we used a linear mixed effects modeling approach using the 'lme4'[71] R package with fixed and random effects to conduct our SOC carbon stock spatial prediction. Linear mixed effect models were used to specify the fixed effect as the WIP probability metric for our primary covariate for SOC. We also investigated multiple remote sensing metrics such as Normalized Difference Vegetation Index (NDVI), Enhanced Vegetation Index (EVI), and the Modified Normalized Difference Water Index (MNDWI), as well as single band reflectance from Landsat imagery as additional fixed effects in the model. We used surficial geology of the HRW to the map riverine quaternary sediments which represent river floodplains that are strongly predicted as wetland areas in the WIP tool but do not develop soil or accumulate organic matter due to recent river water scouring. Surficial geology data were downloaded as 1:100,000 scale polygons from the Washington State Department of Natural Resources geologic information portal. Four broad classes of lithological material and geologic age were extracted from the surficial geology data to provide grouping for SOC samples: Clastic, Glacial Drift, Till/Outwash, and Alluvium. Surficial geology was designated as a random effect due to the uneven sample sizes in each surficial geology group and to account for variation by adjusting the intercept based on surficial geology categories within the model. Surficial geology could be designated as a fixed effect due to fewer than 5 levels (4) which is a common cutoff for random effects, but this has been shown to not affect model performance especially when another parameter is of interest. Choosing between fixed and random effect designation is not straightforward but designating random effects can be helpful to improve parameter estimation such as the WIP in this study[72].

Stepwise variable selection using Akaike information criterion (AIC) was used to determine fixed effect covariates in addition to the WIP tool probability metric and surficial geology random effect. However, there were no significant effects from adding remote sensing metrics to the WIP and surficial geology. Further, the heterogeneity of the forested landscape due to forest harvest was prohibitive for using spectral remote sensing metrics or lidar metrics of forest structure to predict SOC which could weight clearings or reflective surfaces inappropriately in SOC predictive modeling. Additional terrain metrics were also excluded to avoid intercorrelations with the WIP probability covariate which already incorporates terrain information and surficial geology. Overall, the best model according to AIC was also simplest using just the WIP probability with surficial geology classes (Eq. 2).

$$\sqrt{(SOC\ Stock)}_{ij} = \mathcal{X}\beta_{WIP_i} + \mathbf{Z}\alpha_{Surficial\ Geology_j} + {}_{ij} \tag{2}$$

Where $\mathcal{X}$ is the fixed effects design matrix for the $\beta_{WIPi}$ in pedon $i$, $\mathbf{Z}$ is the random effects design matrix for the random effects $\alpha_{Surfical\ Geology_j}$ for an geology type $j$, and $_{ij}$ is our model error described as $_{ij} \sim N(0, \sigma_{\in}^2)$. Pedons sampled from the random effect $\alpha_{Surfical\ Geology_j}$, described as $\alpha_{Surfical\ Geology_j} \sim N(0, \sigma_{\alpha}^2)$ are considered a random sample from a separate normal distribution for each surficial geology type $j$.

## SOC stock prediction

The Eq. 2 model was used to predict SOC stock at a 1 m and 30 cm depth with a $R^2$ of 0.63 and 0.61, respectively, between observed (Pedon SOC stocks) and predicted SOC stock values. The WIP variable as a strongly significant predictor with an estimated non-transformed coefficient for 1 m SOC stock of 391 (95% Conf. Interval: 241–516 MgC ha$^{-1}$, see Supplementary Table 1 for full model parameter estimates and details). This was much higher than the surficial geology random effect (Supplementary Table 1) showing that the finer scale WIP parameter was driving SOC more than the coarser scale patterns of surficial geology. The Root Mean Square Error (RMSE) for the 1 m model was 96.8 MgC ha$^{-1}$ and 31.0 MgC ha$^{-1}$ for the 30 cm model. A leave one out cross validation computed a cross-validation RMSE of 22.8 MgC ha$^{-1}$ for 1 m SOC stocks and 11.7 MgC ha$^{-1}$ for 30 cm SOC stocks. Bootstrapped model predictions for the 1 m model showed 95% confidence intervals (2.5% to 97.5%) around the mean based on 1000 simulations were 216–511 MgC ha$^{-1}$ for the WIP, 3.67–129 MgC ha$^{-1}$ for the intercept, 77.0–131 MgC ha$^{-1}$ for the variance, and 49.5–145 MgC ha$^{-1}$ for the surficial geology random effect intercept. Bootstrapped model predictions for the 30 cm model, were 51.8–147 MgC ha$^{-1}$ for the WIP, 36.7–76.1 MgC ha$^{-1}$ for the intercept, 24.4–41.4 MgC ha$^{-1}$ for the variance, and 24.3–55.8 MgC ha$^{-1}$ for the surficial geology random effect intercept. We note these bootstrapped confidence intervals were computed on the non-transformed model which potentially widens the confidence intervals but allows for better interpretation with results in SOC response variable units of MgC ha$^{-1}$.

All prediction mapping analyses were conducted with R software (version 4.3.0) with the 'terra'[73] package and all map figures were created using Esri ArcGIS Pro (version 3.2.1). Rasters data layers for the WIP probability and surficial geology were projected to the NAD83 UTM Zone 10 (EPSG:26910) and resampled to match the WIP original 4 m pixel resolution of the digital elevation model. SOC stocks at 30 cm and 1 m depths were predicted across the HRW using the two raster data layers and the model from Eq. 2 which resulted a spatially continuous map of the square root SOC that was then back transformed with squaring to result in SOC stocks in MgC ha$^{-1}$. We masked surface water presence by using the median modified normalized difference water index (MNDWI) across a 5-year period from 2016 to 2021. We examined the riverine classification and classified all MNDWI values above 0.30 as river surface water to be masked out. The masking process also removed a small lake located in the mountains on the eastern portion of the watershed and small gravel pits in the center of the watershed. The resulting SOC prediction map was used to calculate the total HRW SOC stock, wetland SOC stock, forested wetland SOC stock, riverine wetland SOC stock, palustrine wetland SOC stock, and non-wetland/upland SOC stock. Wetland SOC stock was estimated by classifying pixels as wetlands with WIP probability ≥50% and we refer back to ref. 33 for the discussion of error with this threshold. We note that this WIP-based classification reflects potential wetland extent but is not meant to confer jurisdictional wetland extent which requires ground truth delineations. Surficial geology delineation of the Hoh River main channel and floodplain was used to classify riverine wetland and palustrine wetland SOC stocks. Forested wetland SOC stocks were estimated from a forest/non-forest mask of wetland SOC stocks derived from tree cover ≥50% in the Global Tree Cover product in ref. 74. Non-wetlands were delineated as the total area

outside of the WIP probability ≥50% and we classify this area and SOC stock as uplands.

We quantified uncertainty using several methods. First, we examined the $R^2$ value of the fit vs. the predicted values in the final model output to judge the overall fit of the model on the actual SOC values in the current dataset. Next we calculated confidence intervals using the 'confint.merMod' function in the lme4 R package[70]. Next, we generated a prediction interval for the model using the 'predictInterval' function in the 'merTools'[75] R package. This function computes a simulated distribution for all parameters in the model. For the random effect simulation, the distribution is simulated by sampling from a multivariate normal distribution defined by the best linear unbiased prediction estimate and the variance-covariance matrix for each level of the grouping terms. The result is a matrix of simulated values for the linear mixed effects model and each random effect grouping term has a matrix for each observation. The 5th and 95th percentiles of the final simulated distribution were used to define the uncertainty in the prediction and root mean square error was calculated from the difference in the fit vs. the predicted values. Finally, we calculated the mapped SOC prediction uncertainty with a bootstrapping approach. Bootstrapped datasets were constructed by sampling the pedon SOC values from the current dataset with replacement, then integrating that dataset into the prediction model in Eq. 1 which was used to further predict SOC across the HRW. In total we used 300 bootstrapped SOC prediction maps where each pixel contained 300 predictions to simulate a distribution from which we extracted the standard deviation to represent the prediction interval uncertainty. We then compared the WIP wetland SOC stocks with 1 m SOC stocks derived from ref. 11 which harmonized the National Wetland Condition Assessment (NWCA) and U.S. Department of Agriculture's National Cooperative Soil Survey (NCSS) Soil Survey Geographic Database (SSURGO) datasets then upscaled the harmonized dataset to spatially explicit inland wetland extent measured by the Landsat-derived National Land Cover Database (NLCD)[11]. We term the spatially explicit maps of wetland SOC from ref. 11 as NWCA-SSURGO which provides the latest mapped wetland SOC stocks at 30 cm and 1 m depths for the continental U.S. at a 30 m resolution. To identify differences between our pedon collection upscaled with WIP and currently mapped estimates, which we term 'cryptic carbon', NWCA-SSURGO SOC stocks were subtracted from the WIP wetland SOC stocks. Standard deviations are reported from ref. 11 for the mapped NWCA-SSURGO SOC estimates and we provide those in Table 2. While NWCA-SSURGO dataset contains NLCD-defined forested wetlands, we instead relied on the >50% forest cover from the Global Tree Cover in ref. 72 for consistency with our estimates of dense canopy forested wetlands within the WIP.

## Wetland size distribution

Wetland size and extent were derived defining wetlands from the WIP tool probability greater than 50%. All wetland pixels greater than 50% were classified as wetland and converted to polygons using the 'terra'[72] and 'sf'[76] R packages. The wetland polygons were filtered to remove all wetlands below 64 m$^2$ which is the area equivalent to 2x2 pixels in order to conservatively estimate wetland size classes. Examination of the wetlands below 64 m$^2$ did not reveal significant cumulative proportions of SOC or extent. Wetlands above 64 m$^2$ were used to extract SOC values in MgC ha$^{-1}$ from the prediction raster. Size classes were defined as quantiles: 1%, 25%, 50%, 75%, 96.4% and 100%, and cumulative sums for SOC and areal extent were calculated. The 96.4% quantile marks the 1-acre or 0.40 ha extent that is the minimum mapping unit of the National Wetland Inventory (NWI), separate from the NLCD, and is used as a threshold for small wetlands in this study. The NWI defines 0.40 ha as the minimum mapping unit due to constraints with manual aerial photo

interpretation but wetlands can still be mapped at smaller extents with less consistently[12].

## Data availability

The soil organic carbon stocks data generated in this study have been deposited in the ORNL DAAC database available at https://daac.ornl.gov/cgi-bin/dsviewer.pl?ds_id=2249 and on Github at https://github.com/ajs0428/CrypticCarbon. The processed mapped data are also available at https://daac.ornl.gov/cgi-bin/dsviewer.pl?ds_id=2249.

## Code availability

All code used in data processing, modeling, mapping, and graphing are available at: https://github.com/ajs0428/CrypticCarbon and https://doi.org/10.5281/zenodo.10426214.

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

## Acknowledgements

We thank Claire Johnson, Hazel Sanders, Abby Nesper, Megan Sayasemawong, and Thomas Kakatsakis for their help with field sampling and sample processing. We thank Maureen Duane for coordination and project management. We thank Frances Biles for discussion on geospatial data collection. We thank the staff at the Olympic National Resources Center for housing accommodations. We thank Jamie Bass and Nicole Kerr at the Nature Conservancy for allowing access to private lands. We thank Amy Yahnke and Amanda Nahlik for discussion and advice on measuring, analyzing, and evaluating wetland carbon stocks. We thank Dongsen Xue for soil analysis at the University of Washington Analytical Service Center. This work was supported by National

Aeronautics and Space Administration Grant Number 80NSSC20K0427 P00003 and USDA Forest Service Joint Venture Agreement 20-JV-11261933-058 AM003. We also acknowledge this work takes place on the traditional lands of the ChalAt'i'lo t'sikAti, Quinault, Quileute, Suquamish, Stillaguamish, and Duwamish tribes.

## Author contributions

L.M.M. conceived the study and assisted with model analysis. M.H. designed the study, performed modeling analysis with the WIP tool, and contributed data analysis on SOC mapping with contributions from C.B. Study development, sample collection strategy, and data analysis were performed by D.B. and D.V.D. The sample collection, laboratory analysis, and SOC mapping with contributions were performed by A.J.S. with help from L.M.M., M.H. and C.B.; A.J.S. wrote the manuscript with contributions from all authors to manuscript writing and interpretation of results.

## Competing interests

The authors declare no competing interests.
