## [Peer Review File · Nature Communications]

Revealing the hidden carbon in forested wetland soilsREVIEWER COMMENTS

Reviewer #1 (Remarks to the Author):

This paper provides a clear description of a novel mapping technique to create continuous gradient of SOC values across the terrestrial aquatic continuum within the Hoh River Watershed. Although the technique that the paper utilizes is novel, it is well couched within prior studies of regional and continental studies that lend needed context to their reported values. This exploration into “cryptic carbon”, wetland SOC that is neglected in current extent projections, is a blind spot in current estimation and could be extrapolated upon in a number of diverse settings and environments. The authors provide robust reasoning for the design and application of this study providing necessary error estimates throughout the manuscript. Additionally, the discussion section of the manuscript successfully extrapolates the implications of this project without overstepping into grandiose hypotheticals. Overall, this paper was pleasant and easy to read, provided a novel investigation into a critical blind spot in carbon accounting, and supplied a framework to explore similarly cryptic carbon in other locations. I only have a few specific criticisms of the paper: (1) the abstract is severely underwritten and misrepresents the novelty of this work and (2) certain elements of field sampling are unclear, as well as the below line-item edits.

Abstract: In general, the abstract is difficult to read without some specific description of the geographical reference and spatial extent of the study as well as the other numbers presented in this abstract. For example, L14-L16 you state that the total catchment SOC was comparable to global SOC maps, but then immediately state that your approach contained 1.7 TgC compared to 0.3 TgC. This seems like an immediate contradiction to the previous statement. This confusion could be the result of missing information regarding where these numbers are derived from in terms of geography and extent. The abstract currently lacks accessibility without reading the rest of the paper. Finally, including some extrapolation, implication, or inference that you explore in the discussion could make this abstract much more enticing.

L13-14 Add some specificity of location and the extent of this catchment. Without some spatial and geographical context the following numbers could be global, continental, or regional.

L17 I would remove “or 383% higher” from this sentence to lend impact to the “483%” in the following sentence

L25 add global before “total soil organic carbon stock”

L42 Add some estimates of freshwater inland wetland size as well as total wetland area in parenthetical statements. Simply saying “most” is too vague. I think the Nahlik and Fennessey paper says about 95% of wetlands are inland freshwater wetlands

L97 Some reference is needed to a paper or dataset

L114 Suggest starting a new paragraph

L137 A 4m vertical or horizontal resolution? Include both in this section

L139 Missing a close parenthesis

L169-171 Why were so many more upland sites sampled than wetland sites when the goal of the study is to explore the SOC in wetlands? I understand that the landscape may be dominated by upland area, but if the goal of this study is to examine SOC in cryptic wetlands, why is the sample size low for these

landcover types? Including a figure of the probability distribution and bins in the supplementary materials may help with this question. Additionally, you write on L165 that you have 1 pedon per bin (30) with 12 additional pedons between the highest and lowest for a total of 42 pedons. However, you only sampled from 36 sites. Were some of these pedons sampled at the same site? Adding sampling locations to Figure 1 could be helpful.

L172 Write out how many riverine and palustrine wetland types were measured ($n=x$). It is unclear what your sample size are for these wetland types, which is especially important in capturing the variance in SOC stocks and in scaling these results to the catchment area.

L187-192 You could reduce the wordiness of this section

L235 Move the unit after the second value instead of the first

L308-309 and L310-311 and L321-322 This seems like a redundant way to talk about SOC distribution over the depth of pedon. Instead, report percentages in depth ranges (0-30 cm, 30-60 cm, 60-90 cm, etc.). The point that nearly 100% of SOC was in the top 1 meter is captured explicitly in later sentences so no need to do it here as well.

L319 Change to "a mean" here and at multiple points in the results

L362-384 and L400-413 Much of these paragraphs seem to fit more into the discussion section comparing different estimates of SOC rather than in the results section. I suggest reducing the amount of text in this section.

L416 Are rivers and lakes masked out of Figure 41 and 4b? It's unclear if this is consistent across the images

L424 "that" needed before wet

L434-436 Reduce the wordiness of this sentence. You essentially repeat the beginning of this sentence again at the end

L444 This is a great section that compares SOC mapping studies, but I recommend splicing this large chunk of text into 2-3 paragraphs (For example at L466, L471).

L485-487 This sentence is possibly too vague. I suggest reeling back the language a little bit. There are many unique landscape features within the Hoh River Catchment that are not replicated broadly outside of this area

L493 Need in-text citation after authors' names

L484-507 This is an interesting idea and the authors do a good job of not overemphasizing their results by mentioning the various limitations of this exercise. This is an interesting inclusion to the paper!

L508-544 This is also a thought provoking section that extends the findings of the paper. Would it be possible to discuss that your estimations of wetland extent are based on maps from 2012 and 2013 and some of the possible changes that have occurred in the study area since those maps were made? Just as an additional point to this already stellar section

Reviewer #2 (Remarks to the Author):

Stewart et al. present an interesting geospatial analysis investigating the implications of wetland mapping for the estimation of soil organic carbon stocks in notoriously difficult to map forested wetlands. They collected soil cores from 36 locations across a watershed in the Pacific Northwest, with sampling locations informed by a probabilistic map of wetland occurrence. Sieved cores were analyzed for SOC content, and the variability in SOC content was evaluated with a mixed effect model. The model was then used to estimate SOC across the entire watershed and those results were evaluated according to different wetland classes and relative to other datasets on wetland carbon.

The results suggest that there is a substantial amount of SOC stored in this region's forested wetlands that would be overlooked and unaccounted for without the improved wetland mapping approach that includes forested wetlands. While it is a promising study that addresses a challenging and important problem, there are several aspects of the manuscript that could be improved for clarity and completeness. However as currently presented, there is not a very clear line of inference between the results and all the conclusions, and many of the implications and speculation in the discussion seem to extend beyond the nature of the results presented. The paper would be much stronger with more detail regarding assumptions and limitations. For example, while the discussion includes a nice point about limitations in applying the methodology to different regions, it may be more realistic to use 95% confidence interval ranges in the discussion when using SOC stocks extrapolated from model results.

Areas for improvement:

1. More details should be reported about the results of the statistical model, such as including the model coefficients not just the form of equation 2. Readers would benefit from discussion about the relative importance of WIP metric and surficial geology in the model, to help draw a clearer line of inference from the statistical analysis to the conclusion that WIP is a significant covariate. It would be somewhat difficult to recreate the stepwise model selection procedure used because the remote sensing and terrain metrics tested do not appear to be comprehensively provided and only the variables of the final model are given. Consider using different symbol shapes for the different geology types in Figure 2, since the number of samples within each geology type is uneven and the strength and form of the relationship between WIP and SOC varies (only evident in the supplemental data).
2. Intra-wetland variability in carbon stocks should be addressed. As there can be a high degree of variability in SOC stocks within wetlands (eg. Stewart et al 2023 ERL, 10.1088/1748-9326/acd26a; Tangen and Bansal 2020 10.1016/j.scitotenv.2020.141444; Pearse et al 2017 10.1002/Ino.10735), it would be helpful to include details in the methodology about how the specific sampling sites were chosen, and/or the extent to which the WIP mapping is intended to account for such variability.
3. I'm curious whether the authors considered measuring carbon content bound up in macroaggregates in the greater than 2mm fraction. Is there any information about SOC by size fraction for wetlands in this region, to know how much may be underestimated?
4. As true with most discussions of wetland mapping, it would be helpful to be more consistent and obvious when describing different datasets and the baselines used for making comparisons. For example the WIP tool is trained on the NWI polygons but also designed to identify areas missing in "most wetland inventories" – is the assumption here that the NWI is accurate but incomplete? How well does the NWI represent wetlands without standing water or under tree canopies in this region? It may be more clear to consistently refer to the Uhan et al. dataset as "NWCA-derived" instead of just NWCA. It would also

help to more clearly explain the wetland categorization used because forested wetlands are a subset of palustrine wetlands in the NWI classification however there is a different definition used here.

5. The discussion section would be much stronger with more detail and interpretation of the strengths and weaknesses of the statistical model, such as why the model is underestimating and overestimating SOC for different sampling sites. The discussion of vulnerability to anthropogenic disturbance seems unconnected to the themes presented in the introduction and only loosely connected to the extrapolated model results. At the very least the extrapolated values should be presented as 95% confidence ranges rather than mean values.

Reviewer #3 (Remarks to the Author):

The paper addresses an important topic, and gap in our understanding of the extent of forested wetlands and the carbon they store. The approach using probabilistic wetland mapping is shown to be effective in bettering estimates of the extent and distribution of forested wetlands in the HRW, and this, combined with field measures of soil carbon, provide a better estimate of SOC stocks.

The format of the paper is not standard for this journal, but I assume this will be corrected if it is published.

The paper relies heavily on the methods described in Halabisky et al., and I recommend that the authors include more details on the methods used in this study in the supplementary materials.

I would like to see some discussion of the limitations of this approach since only 1 watershed was used to make these estimates. This is a small portion of the total US – how would this work in other types of terrain, etc.

In total, 8 wetlands were sampled compared to 28 uplands – can an explanation be offered on why the wetland sample size was so much smaller?

Line 323 – does this mean wetlands were included on the NWI and fell in the WIP range between 25-50? This isn't clear.

Line 350 – using the SOC: ecosystem extent ratios to make these comparisons is quite effective.

Table 2 – it would help to clarify that the 3 wetland types are subsets of the Wetland category overall, and are not additional to it.

Table 2 and discussion, e.g., lines 490-498:

The comparison of the results of this paper and the NWCA are not correct- the NWCA did not map any

wetlands as implied on Table 2, rather it relied on existing maps (in this case, maps used in the USFWS Status and Trends plots (S&T) as part of the NWI) to select sites for field sampling. The study design in the NWCA then uses the field data and the relationship of the sampled site to the population at large, to make estimates for the larger population, in this case for SOC. Table 2 shows the NWCA as a source of mapping data which is incorrect (it is specified on line 493), this should be corrected to S&T or NWI. In the Ufran et al. paper, the NWCA data was combined with SSURGO spatial soil data to make estimates, so the source of the mapping is SSURGO. Nahlik and Fennessy used the study design of the NWCA and S&T maps to scale up estimates of SOC collected at field sites randomly selected from the S&T plots. So this study relied on existing maps, which suffer from the same underestimate of forested wetlands that the authors are addressing (and which has long been known). This is more of a programmatic limitation for efforts such as the NWCA because the NWI is the only source of maps on wetland extent available nationally. Clearly the limitations of mapping are an issue for large scale efforts to report on wetland condition or services so it may be more accurate to say that the maps are the issue, not the way they were used – which is why this study was devised in in the first place.

Finally, the NWCA did not select 0.5 ha because it is a minimum mapping unit, this was the agreed size of the assessment area (AA) used in the NWCA based on earlier studies. Other notes on this section: Fennessy is misspelled, and Nahlik et al. (line 504) is not in the references.

Cover Letter

Dear Reviewers,

We want to thank you for your time and feedback to our manuscript which improved the original submission. We want to note that during the revision process we found minor errors in the analysis which we have rectified and that have not changed the overall results or conclusions in the manuscript. The main error was a missing data entry for the rock percentage for a pedon sample. After correction, we have reanalyzed the dataset, models, and reproduced all figures in the manuscript. The results contain some slightly different numbers but magnitude of changes are not significant, e.g. Mean pedon SOC stock in the HRW was $221 \pm 27 \text{ MgC ha}^{-1}$ which was corrected to $226 \text{ MgC} \pm 27 \text{ ha}^{-1}$. For mapping, Total HRW Mean SOC stock was $74.2 \pm 16 \text{ MgC ha}^{-1}$ which was corrected to $72.9 \pm 12.5 \text{ MgC ha}^{-1}$. We also note minor corrections below:

- Table 1. Contains corrected values for the 60 cm depth which was originally mistakenly calculated at 90 cm.
- Table 1. Riverine wetlands were also corrected and recalculated to only include $\text{WIP} \geq 50\%$. Originally some $\text{WIP} < 50\%$ pedons were accidentally included in the Riverine wetland calculation.
- We changed the 64.2 PgC cited in Gautham et al., 2022 for 30 cm depth SOC stocks to 65 PgC for 1 m depth SOC stock estimates reported in Lajtha et al., 2018 SOCCR in order to compare 1 m estimates across multiple sources.

We encourage any review and comparisons to the previously submitted manuscript with the note that we moved the Methods section to the end to reflect the formatting of Nature Communications. We also have uploaded code used in the analysis to <https://github.com/ajs0428/CrypticCarbon> and encourage reviews there as well.

Another note is that the paper describing the WIP tool development has been published in Hydrology and Earth System Sciences at <https://doi.org/10.5194/hess-27-3687-2023> and replaces the pre-print version in the original submission.

The revised manuscript was greatly improved by the thorough re-analysis and was facilitated by reviewer comments and revision suggestions. We greatly appreciate this opportunity to present more accurate scientific reporting.

Sincerely,
Anthony J. Stewart
Meghan Halabisky
Chad Babcock
David Butman
Dave D'Amore
L. Monika Moskal

Reviewer Comments

Reviewer #1 (Remarks to the Author):

- This paper provides a clear description of a novel mapping technique to create continuous gradient of SOC values across the terrestrial aquatic continuum within the Hoh River Watershed. Although the technique that the paper utilizes is novel, it is well couched within prior studies of regional and continental studies that lend needed context to their reported values. This exploration into “cryptic carbon”, wetland SOC that is neglected in current extent projections, is a blind spot in current estimation and could be extrapolated upon in a number of diverse settings and environments. The authors provide robust reasoning for the design and application of this study providing necessary error estimates throughout the manuscript. Additionally, the discussion section of the manuscript successfully extrapolates the implications of this project without overstepping into grandiose hypotheticals. Overall, this paper was pleasant and easy to read, provided a novel investigation into a critical blind spot in carbon accounting, and supplied a framework to explore similarly cryptic carbon in other locations. I only have a few specific criticisms of the paper: (1) the abstract is severely underwritten and misrepresents the novelty of this work and (2) certain elements of field sampling are unclear, as well as the below line-item edits.
 - o We greatly appreciate the review and suggested revisions and have made several updates to the manuscript that address the two points. Several updates are noted below
- Abstract: In general, the abstract is difficult to read without some specific description of the geographical reference and spatial extent of the study as well as the other numbers presented in this abstract. For example, L14-L16 you state that the total catchment SOC was comparable to global SOC maps, but then immediately state that your approach contained 1.7 TgC compared to 0.3 TgC. This seems like an immediate contradiction to the previous statement. This confusion could be the result of missing information regarding where these numbers are derived from in terms of geography and extent. The abstract currently lacks accessibility without reading the rest of the paper. Finally, including some extrapolation, implication, or inference that you explore in the discussion could make this abstract much more enticing.
 - o We have revised the abstract to include geographical information and more context on the datasets being compared. This should help clarify the wetland SOC comparison mentioned above. We see that the abstract length for this publication is expected to be 150 words and we hope the editor can accept the revisions here although it lengthens the abstract.
- L13-14 Add some specificity of location and the extent of this catchment. Without some spatial and geographical context the following numbers could be global, continental, or regional.
 - o We added in characteristics and names for the HRW catchment and clarification on global estimates
- L17 I would remove “or 383% higher” from this sentence to lend impact to the “483%” in the following sentence
 - o Agreed and removed

- L25 add global before “total soil organic carbon stock”
 - o Added
- L42 Add some estimates of freshwater inland wetland size as well as total wetland area in parenthetical statements. Simply saying “most” is too vague. I think the Nahlik and Fennessey paper says about 95% of wetlands are inland freshwater wetlands
 - o Indeed, 95% of the total wetland area is freshwater wetland and Nahlik and Fennessey, 2016 cited Dahl, 2011 as a source. We include the latter within the citation
- L97 Some reference is needed to a paper or dataset
 - o Edited to reflect statistics come from reference 35 (Halabisky et al., 2023) which has been published recently in HESS.
- L114 Suggest starting a new paragraph
 - o Indent added
- L137 A 4m vertical or horizontal resolution? Include both in this section
 - o We added “...4 m cell size resolution...” to specify the length and width of the raster cells
- L139 Missing a close parenthesis
 - o Fixed
- L169-171 Why were so many more upland sites sampled than wetland sites when the goal of the study is to explore the SOC in wetlands? I understand that the landscape may be dominated by upland area, but if the goal of this study is to examine SOC in cryptic wetlands, why is the sample size low for these landcover types? Including a figure of the probability distribution and bins in the supplementary materials may help with this question. Additionally, you write on L165 that you have 1 pedon per bin (30) with 12 additional pedons between the highest and lowest for a total of 42 pedons. However, you only sampled from 36 sites. Were some of these pedons sampled at the same site? Adding sampling locations to Figure 1 could be helpful.
 - o Yes, part of the goal is to examine SOC in cryptic wetlands but also compare and contrast with the surrounding landscape (uplands). In order to do so, we felt we needed to provide unbiased sampling across the entire HRW. Our original stratification reflected this by trying to sample evenly across the distribution. But we faced difficulty navigating to those exact pixels in the backcountry, especially with a fine resolution map and limited GPS navigational equipment. We tried to not project a bias towards wetland or non-wetland at each location for the 30 original sample points. Once completed, we utilized additional field time to obtain 6 additional samples which were split (3 each) between the upper and lower ends of the WIP distribution to bracket the ends our pedon dataset. This resulted in many more samples located in the lower-middle part of the WIP distribution in our sample pedons which were original chosen as upper-middle part of the WIP distribution. While this does weight our samples towards uplands, we note similarly to the reviewer that this somewhat reflects the overall landscape WIP distribution and supports the modeling the entire HRW based on these pedons. We have provided additional text explaining the sampling procedure and supplementary figures to show the sample locations with WIP values (Supplementary Figure 2), the overall WIP model distribution in the HRW

(Supplementary Figure 3a), and the pedon sample distribution (Supplementary Figure 3b).

- L172 Write out how many riverine and palustrine wetland types were measured (n=x). It is unclear what your sample size are for these wetland types, which is especially important in capturing the variance in SOC stocks and in scaling these results to the catchment area.
 - o Added in parentheses
- L187-192 You could reduce the wordiness of this section
 - o Removed loss on ignition since only the CHN values are reported. Wording is shifted around to be more straightforward
- L235 Move the unit after the second value instead of the first
 - o Fixed
- L308-309 and L310-311 and L321-322 This seems like a redundant way to talk about SOC distribution over the depth of pedon. Instead, report percentages in depth ranges (0-30 cm, 30-60 cm, 60-90 cm, etc.). The point that nearly 100% of SOC was in the top 1 meter is captured explicitly in later sentences so no need to do it here as well.
 - o We rephrased to match the reviewer comment and show percentages within the horizons depth intervals. We also changed this for the other land types in the rest of the section
- L319 Change to “a mean” here and at multiple points in the results
 - o Fixed
- L362-384 and L400-413 Much of these paragraphs seem to fit more into the discussion section comparing different estimates of SOC rather than in the results section. I suggest reducing the amount of text in this section.
 - o We moved the last part of the section containing “Most of the cryptic carbon stock is due to the 273% increase in potential wetland extent from the WIP \geq 50%, However, new wetland extent contained wetlands with a higher mean SOC stock of 259 ± 72 (187-331) MgC ha⁻¹ compared to the mean SOC stock in NWCA wetlands (184 ± 108 (76-292) MgC ha⁻¹) showing a new inclusion of wetlands with high SOC stocks.” To the discussion section. But we feel that the section on wetland size distribution should remain in the results
- L416 Are rivers and lakes masked out of Figure 41 and 4b? It’s unclear if this is consistent across the images
 - o Added text to the caption to indicate consistent masking
- L424 “that” needed before wet
 - o Added
- L434-436 Reduce the wordiness of this sentence. You essentially repeat the beginning of this sentence again at the end
 - o Removed the latter part of the sentence
- L444 This is a great section that compares SOC mapping studies, but I recommend splicing this large chunk of text into 2-3 paragraphs (For example at L466, L471).
 - o We separated at these points to make new paragraphs at L308
- L485-487 This sentence is possibly too vague. I suggest reeling back the language a little bit. There are many unique landscape features within the Hoh River Catchment that are not replicated broadly outside of this area

- We reduced vague elements here and referenced the limitations of extrapolation with the HRW
- L493 Need in-text citation after authors' names
 - Fixed
- L484-507 This is an interesting idea and the authors do a good job of not overemphasizing their results by mentioning the various limitations of this exercise. This is an interesting inclusion to the paper!
 - We very much appreciate the comment here!
- L508-544 This is also a thought provoking section that extends the findings of the paper. Would it be possible to discuss that your estimations of wetland extent are based on maps from 2012 and 2013 and some of the possible changes that have occurred in the study area since those maps were made? Just as an additional point to this already stellar section
 - The comment is very much appreciated again and we have added text at L388-391 to reflect potential improvements to the current study based on lidar maps

Reviewer #2 (Remarks to the Author):

- Stewart et al. present an interesting geospatial analysis investigating the implications of wetland mapping for the estimation of soil organic carbon stocks in notoriously difficult to map forested wetlands. They collected soil cores from 36 locations across a watershed in the Pacific Northwest, with sampling locations informed by a probabilistic map of wetland occurrence. Sieved cores were analyzed for SOC content, and the variability in SOC content was evaluated with a mixed effect model. The model was then used to estimate SOC across the entire watershed and those results were evaluated according to different wetland classes and relative to other datasets on wetland carbon.

The results suggest that there is a substantial amount of SOC stored in this region's forested wetlands that would be overlooked and unaccounted for without the improved wetland mapping approach that includes forested wetlands. While it is a promising study that addresses a challenging and important problem, there are several aspects of the manuscript that could be improved for clarity and completeness. However as currently presented, there is not a very clear line of inference between the results and all the conclusions, and many of the implications and speculation in the discussion seem to extend beyond the nature of the results presented. The paper would be much stronger with more detail regarding assumptions and limitations. For example, while the discussion includes a nice point about limitations in applying the methodology to different regions, it may be more realistic to use 95% confidence interval ranges in the discussion when using SOC stocks extrapolated from model results.

Areas for improvement:

- 1. More details should be reported about the results of the statistical model, such as including the model coefficients not just the form of equation
 - We added more detail for our choices in the model structure in the methods section: SOC stock modeling and covariates. We also added the untransformed estimated coefficient for the WIP in the Model predictions and mapping of SOC stocks section of the results and a table of the full model parameter estimates in

the Supplementary Table 2 which includes bootstrapped 95% confidence intervals.

- 2. Readers would benefit from discussion about the relative importance of WIP metric and surficial geology in the model, to help draw a clearer line of inference from the statistical analysis to the conclusion that WIP is a significant covariate. It would be somewhat difficult to recreate the stepwise model selection procedure used because the remote sensing and terrain metrics tested do not appear to be comprehensively provided and only the variables of the final model are given. Consider using different symbol shapes for the different geology types in Figure 2, since the number of samples within each geology type is uneven and the strength and form of the relationship between WIP and SOC varies (only evident in the supplemental data).
 - o More text was added to the methods in the sections “SOC stock modeling and covariates” and “SOC Stock Prediction” as well as an additional table of model parameter estimates (Supplementary Table 1.) to explain the model parameters and their estimates more comprehensively. The graph in Figure 1, (Previously Figure 2) was also amended to reflect the differences in the surficial geology categories with different shapes. A note in the results directs readers here on L183-L184

- 2. Intra-wetland variability in carbon stocks should be addressed. As there can be a high degree of variability in SOC stocks within wetlands (eg. Stewart et al 2023 ERL, 10.1088/1748-9326/acd26a; Tangen and Bansal 2020 10.1016/j.scitotenv.2020.141444; Pearse et al 2017 10.1002/lno.10735), it would be helpful to include details in the methodology about how the specific sampling sites were chosen, and/or the extent to which the WIP mapping is intended to account for such variability.
 - o Indeed, more discussion on intra-wetland variability is warranted and we have added a new paragraph starting at L305-307 utilizing the sources listed here.
 - o We also added descriptions of the sample collection into the “Field sampling” section that describes more detail into how sample sites were collected.

- 3. I’m curious whether the authors considered measuring carbon content bound up in macroaggregates in the greater than 2mm fraction. Is there any information about SOC by size fraction for wetlands in this region, to know how much may be underestimated?
 - o We did not consider measuring the carbon content in macroaggregates and instead focused on the <2mm fraction which is reported for most SOC stocks. It also may be more complex to answer in the current study. However, this is an interesting avenue for future research involving the quality of SOC especially mineral associated organic matter. We are also not aware of details regarding this fraction for wetlands in this region.

- 4. As true with most discussions of wetland mapping, it would be helpful to be more consistent and obvious when describing different datasets and the baselines used for making comparisons. For example the WIP tool is trained on the NWI polygons but also designed to identify areas missing in “most wetland inventories” – is the assumption here that the NWI is accurate but incomplete? How well does the NWI represent wetlands without standing water or under tree canopies in this region? It may be more clear to consistently refer to the Uhan et al. dataset as “NWCA-derived” instead of just NWCA. It would also help to more clearly explain the wetland categorization used because

forested wetlands are a subset of palustrine wetlands in the NWI classification however there is a different definition used here.

- The assumption that the NWI is accurate but incomplete is correct and details of this mapping are discussed in Halabisky et al., 2023 which is the WIP tool paper. Here, there is substantial evidence that wetlands under tree canopy are often missing. Dahl 2011 also mentions the challenges of mapping forested wetlands in the U.S.
 - We added a sentences explaining the definition of Palustrine used in the study in the results at L110-113 and in the methods at L450-453.
 - We have revised our references to the Ufran dataset as the NWCA-derived dataset.
- 5. The discussion section would be much stronger with more detail and interpretation of the strengths and weaknesses of the statistical model, such as why the model is underestimating and overestimating SOC for different sampling sites. The discussion of vulnerability to anthropogenic disturbance seems unconnected to the themes presented in the introduction and only loosely connected to the extrapolated model results. At the very least the extrapolated values should be presented as 95% confidence ranges rather than mean values.
- We added some discussion of the model limitations in the beginning of the discussion at L239. We also point to the ‘Comparisons to other SOC mapping studies’ section at L262 for some discussion of estimate.
 - We acknowledge that the disturbance aspect is weighted more in the discussion but we feel that additional introduction material of disturbance effects on wetlands and wetland SOC would give readers an impression that the research is evaluating disturbance. However, disturbance is a key facet of wetland SOC and while not analyzed here, it is critical for future study and investigation.
 - Because we are extrapolating SOC stock based on the potential wetland extent and not our own data, we do not have the means to calculate 95% confidence intervals but do provide the estimated standard deviations which are provided in the other studies. However, the reviewer provides a good point that 95% confidence interval ranges should be included. Therefore for our own data we have amended Table 2 and values of estimated SOC stocks throughout the manuscript to include 95% confidence interval estimates.

Reviewer #3 (Remarks to the Author):

- The paper addresses an important topic, and gap in our understanding of the extent of forested wetlands and the carbon they store. The approach using probabilistic wetland mapping is shown to be effective in bettering estimates of the extent and distribution of forested wetlands in the HRW, and this, combined with field measures of soil carbon, provide a better estimate of SOC stocks.
- The format of the paper is not standard for this journal, but I assume this will be corrected if it is published.
 - We moved the methods section to the end to match formatting in Nature Communications

- The paper relies heavily on the methods described in Halabisky et al., and I recommend that the authors include more details on the methods used in this study in the supplementary materials.
 - o We added a section into the supplementary materials that describes the WIP tool and summarizes Halabisky et al., 2023
- I would like to see some discussion of the limitations of this approach since only 1 watershed was used to make these estimates. This is a small portion of the total US – how would this work in other types of terrain, etc.
 - o We expanded the ‘Comparison to other mapping studies’ section with more discussion on limitations particularly at L308.
- In total, 8 wetlands were sampled compared to 28 uplands – can an explanation be offered on why the wetland sample size was so much smaller?
 - o A similar question was posed by Reviewer 1 and we feel the answer above satisfies the question here. We added more discussion and explanation of our site choices into the manuscript and supplementary materials
- Line 323 – does this mean wetlands were included on the NWI and fell in the WIP range between 25-50? This isn’t clear.
 - o We removed the “included as wetlands” phrase. The 25-50% range was supposed to apply just to a mesic category and does not represent wetlands.
- Line 350 – using the SOC: ecosystem extent ratios to make these comparisons is quite effective.
 - o Very much appreciated
- Table 2 – it would help to clarify that the 3 wetland types are subsets of the Wetland category overall, and are not additional to it.
 - o We added asterisks and italicized text to make this clearer for the wetland classes
- Table 2 and discussion, e.g., lines 490-498: The comparison of the results of this paper and the NWCA are not correct- the NWCA did not map any wetlands as implied on Table 2, rather it relied on existing maps (in this case, maps used in the USFWS Status and Trends plots (S&T) as part of the NWI) to select sites for field sampling. The study design in the NWCA then uses the field data and the relationship of the sampled site to the population at large, to make estimates for the larger population, in this case for SOC. Table 2 shows the NWCA as a source of mapping data which is incorrect (it is specified on line 493), this should be corrected to S&T or NWI. In the Uhan et al. paper, the NWCA data was combined with SSURGO spatial soil data to make estimates, so the source of the mapping is SSURGO. Nahlik and Fennessy used the study design of the NWCA and S&T maps to scale up estimates of SOC collected at field sites randomly selected from the S&T plots. So this study relied on existing maps, which suffer from the same underestimate of forested wetlands that the authors are addressing (and which has long been known). This is more of a programmatic limitation for efforts such as the NWCA because the NWI is the only source of maps on wetland extent available nationally. Clearly the limitations of mapping are an issue for large scale efforts to report on wetland condition or services so it may be more accurate to say that the maps are the issue, not the way they were used – which is why this study was devised in the first place.

- We very much appreciate this comment and opportunity to make these distinctions. Rightfully so, the NWCA is meant as a representation of the population of US wetlands, not as a mapping resource. We try to make this explicit that mapping is the main issue at L355-356. Across the results and discussion we revised to emphasize that the upscaling with current wetland maps is the reason for underestimates, particularly at L335-347. In fact, we highly value this as a great opportunity to utilize the NWCA data which is publicly available and have made an effort to include statements reflecting this in the manuscript at L334 and L356.
- Finally, the NWCA did not select 0.5 ha because it is a minimum mapping unit, this was the agreed size of the assessment area (AA) used in the NWCA based on earlier studies. Other notes on this section: Fennessy is misspelled, and Nahlik et al. (line 504) is not in the references.
 - We have made the distinction that 0.5 ha was chosen for NWI mapping and not for NWCA which has an assessment area protocol not related to mapping in the methods.
 - Additional corrections related to spelling originally were made

REVIEWERS' COMMENTS

Reviewer #1 (Remarks to the Author):

I appreciate the author's responses to my earlier questions and comments. I found their responses to all author comments to be holistically successful in satisfying any concerns expressed. I emphatically recommend this study for publication.

Reviewer #2 (Remarks to the Author):

I believe the revised manuscript successfully addresses issues in the previous version, includes necessary methodological details and supporting information, and is now suitable for publication.

Reviewer #3 (Remarks to the Author):

The authors have done good work to address the comments of the reviews. I support publication once the few comments below are addressed.

- Pg 21, line 325 – the authors mention that the HRW is not fully representative of forested regions in the rest of CONUS due to its climate, forest species composition, and land forms. This is an important point and so some additional text about how these differences might affect one's ability to use this approach in other regions is needed. What changes to the approach might be needed for it to be applied elsewhere?

- The text gives a much more clear picture of the data used from SSURGO was leveraged by the NWCA data (e.g., page 21), however Table 2 gives the NWCA as the source of data, which seems incorrect. If the data from the harmonized data from Uhan et al. and SSURGO it seems this should be given here as the source of the data.

Reviewer Comments and Responses

Reviewer #1 (Remarks to the Author):

- I appreciate the author's responses to my earlier questions and comments. I found their responses to all author comments to be holistically successful in satisfying any concerns expressed. I emphatically recommend this study for publication.
 - We greatly appreciated the feedback and review process which improved the manuscript perspective and scope.

Reviewer #2 (Remarks to the Author):

- I believe the revised manuscript successfully addresses issues in the previous version, includes necessary methodological details and supporting information, and is now suitable for publication.
 - We greatly appreciated the reviewer comments that improved our manuscript methodology and discussion.

Reviewer #3 (Remarks to the Author):

- The authors have done good work to address the comments of the reviews. I support publication once the few comments below are addressed.
 - Thank you for the helpful comments which have greatly improved our manuscript during the review.
- Pg 21, line 325 – the authors mention that the HRW is not fully representative of forested regions in the rest of CONUS due to its climate, forest species composition, and land forms. This is an important point and so some additional text about how these differences might affect one's ability to use this approach in other regions is needed. What changes to the approach might be needed for it to be applied elsewhere?
 - We appreciate the comment as this is a good opportunity to add clarification and information about our study location. We added new text at this location in the manuscript that expands on differences between the HRW and other areas in CONUS, noting that further analysis will need to incorporate a variety of parameters to capture cryptic carbon in forested wetlands.
- The text gives a much more clear picture of the data used from SSURGO was leveraged by the NWCA data (e.g., page 21), however Table 2 gives the NWCA as the source of data, which seems incorrect. If the data from the harmonized data from Uhan et al. and SSURGO it seems this should be given here as the source of the data.
 - We agree that it is important to distinguish the data sources clearly and therefore we amended Table 2 NWCA data source and changed all “NWCA-derived” to “NWCA-SSURGO” to reflect the source of the data is a harmonized combination of NWCA and SSURGO. We feel this should provide a better description of the data being used for comparison.